# Performance Evaluation of RapiSure (EDGC) COVID-19 S1 RBD IgG/Neutralizing Ab Test for the Rapid Detection of SARS-CoV-2 Antibodies

**DOI:** 10.3390/diagnostics13040643

**Published:** 2023-02-09

**Authors:** Ha Nui Kim, Jung Yoon, Woong Sik Jang, Chae Seung Lim

**Affiliations:** Department of Laboratory Medicine, Korea University College of Medicine, Seoul 08308, Republic of Korea

**Keywords:** COVID-19 antibodies, neutralizing antibodies, SARS-CoV-2, rapid chromatographic immunoassay

## Abstract

The accurate detection of anti-neutralizing SARS-CoV-2 antibodies can aid in the understanding of the development of protective immunity against COVID-19. This study evaluated the diagnostic performance of the RapiSure (EDGC) COVID-19 S1 RBD IgG/Neutralizing Ab Test. Using the 90% plaque reduction neutralization test (PRNT_90_) as a reference, 200 serum samples collected from 78 COVID-19-positive and 122 COVID-19-negative patients were divided into 76 PRNT_90_-positive and 124 PRNT_90_-negative groups. The ability of the RapiSure test to detect antibodies was compared to that of the STANDARD Q COVID-19 IgM/IgG Plus test and that of PRNT_90_. The positive, negative, and overall percent agreement between the RapiSure and STANDARD Q test was 95.7%, 89.3%, and 91.5%, respectively, with a Cohen’s kappa of 0.82. The RapiSure neutralizing antibody test results revealed a sensitivity of 93.4% and a specificity of 100% compared to the PRNT results, with an overall percent agreement of 97.5% and Cohen’s kappa of 0.95. The diagnostic performance of the RapiSure test was in good agreement with the STANDARD Q COVID-19 IgM/IgG Plus test and comparable to that of the PRNT. The RapiSure S1 RBD IgG/Neutralizing Ab Test was found to be convenient and reliable and, thus, can provide valuable information for rapid clinical decisions during the COVID-19 pandemic.

## 1. Introduction

The coronavirus disease (COVID-19) pandemic still remains a global concern even after 2 years after the first reported case of severe acute respiratory syndrome coronavirus 2 (SARS-CoV-2) infection [1]. Molecular diagnostic tests for SARS-CoV-2 infection and serological tests for the corresponding antibody responses are essential for patient management. Serological tests are especially useful in public health management in terms of pandemic surveillance, assessment of vaccine effectiveness, and evaluation of immune responses [2,3]. 

Humoral and cell-mediated responses are the two arms of the adaptive immune system, which protect against infection and reduce disease severity [4]. Protection is mainly mediated by neutralizing antibodies (nAbs) against SARS-CoV-2 [5]. Almost all patients with COVID-19 develop detectable nAbs after 3–4 weeks of illness [6,7]. Among the virus neutralization assays, which measure serum nAb titers, the plaque reduction neutralization test (PRNT) is considered the gold standard due to its high sensitivity [8]. However, PRNT is technically demanding, time-consuming, and requires biosafety level 3 facilities and trained personnel [9]. Recently, an ELISA-based neutralization assay was developed, but it requires additional equipment, such as a microplate reader [10].

The humoral response to SARS-CoV-2 infection is directed against the viral spike (S) and nucleocapsid proteins [11]. The S protein is responsible for viral entry by interacting with the angiotensin-converting enzyme 2 (ACE2) receptor via a receptor-binding domain (RBD), located in its S1 domain [12,13]. The RBD is the most immunodominant epitope of nAbs [14], and the detection of anti-RBD antibodies is employed in tests, such as point-of-care testing (POCT) [15,16]. Unlike ELISA, POCT does not require sample preparation and is more cost-effective, rapid, and easy to use. Fast and convenient evaluation of SARS-CoV-2 antibodies can aid in determining the level of immune status during the ongoing response to the pandemic. 

In this study, the diagnostic performance of the RapiSure (EDGC) COVID-19 S1 RBD IgG/Neutralizing Ab Test (EDGC, Incheon, Republic of Korea) was evaluated and compared to that of the STANDARD Q COVID-19 IgM/IgG Plus test (SD Biosensor, Suwon, Republic of Korea), the first diagnostic device to detect SARS-CoV-2 antibodies approved by Ministry of Foods and Drug Safety in Korea.

## 2. Materials and Methods 

### 2.1. Sample Collection 

Since the COVID-19 pandemic, COVID-19 RT-PCR tests, including the STANDARD M nCoV Real-Time Detection kit (SD Biosensor, Suwon, Republic of Korea) and Allplex 2019-nCoV Real-time PCR (Seegene, Seoul, Republic of Korea), have been routinely implemented at Korea University Guro Hospital. This study used 200 serum samples from patients who visited the Korea University Guro Hospital with respiratory symptoms from December 2020 to September 2021. Using RT-PCR analysis, the samples were divided into COVID-19-positive (n = 78) and -negative groups (n = 122); for each positive sample, the number of days after illness onset was recorded. The samples were analyzed for the presence of nAbs against SARS-CoV-2, with the PRNT used as a reference method. Before testing, the collected serum samples were stored at −70 °C. This study was approved by the Institutional Review Board of the Korea University Guro Hospital (2021GR0481).

### 2.2. RapiSure COVID-19 S1 RBD IgG/Neutralizing Ab Test 

The RapiSure COVID-19 S1 RBD IgG/Neutralizing Ab Test (RapiSure test) is a rapid, portable qualitative chromatographic colloidal gold-based lateral flow immunoassay that uses two test lanes: one for the detection of S1 RBD IgG antibody and the other for the detection of anti-SARS-CoV-2 nAbs. The test is initiated by adding 25 μL of serum sample to the sample collection sites of the test cassette. In each test lane, the sample then moves chromatographically through a filter pad, a conjugate pad, a nitrocellulose membrane that contains a control (C) band and a test (T) band, and a moisture absorption pad. 

In the S1 RBD IgG test lane, the C and T bands are coated with goat anti-chicken-IgY and mouse monoclonal anti-human IgG antibodies, respectively. If anti-S1 RBD antibodies are present, they form immune complexes with gold-labeled S1 RBD antigen. The complexes then react with the anti-human IgG antibody in the T band and develop a red line.

In the nAb test lane, the presence of anti-SARS-CoV-2 nAbs is indicated by the absence of a colored line in the T band. The SARS-CoV-2 nAbs form immune complexes with gold-labeled S1 protein. These immune complexes cannot interact with ACE2 antigen in the T band, resulting in a weak or absent red line.

The C band in both the test lanes turns from blue to red when the sample passes through the membrane, indicating a valid test. To avoid false results, the results were read 10–15 min after test initiation. A schematic diagram and pictures of the RapiSure test are shown in Figure 1.

### 2.3. Performance Comparison of the RapiSure Test with the PRNT and the STANDARD Q COVID-19 IgM/IgG Plus Test 

The PRNT was used as a reference method to confirm the presence of serum anti-SARS-CoV-2 nAbs. The PRNT was performed as previously described at the Department of Microbiology of Korea University where a biosafety level three facility is available [17]. The highest serum dilution that resulted in 90% (PRNT_90_) and 50% (PRNT_50_) reductions in viral plaque numbers when compared to controls was determined. PRNT_90_ and PRNT_50_ titers that were diluted by at least 1:20 were considered positive for anti-SARS-CoV-2 nAbs [18]. For five positive samples that had a PRNT_90_ titer dilution of 1:160, serial dilutions were used to measure the limit of detection (LoD) in the RapiSure test. Positive PRNT samples were divided into low- and high-titer groups and used to evaluate the performance of the RapiSure test.

Additionally, results with the previously marketed STANDARD Q COVID-19 IgM/IgG Plus test (STANDARD Q test) were compared with those of the RapiSure S1 RBD IgG test. For paired comparisons, only IgG results from the STANDARD Q test were used. The STANDARD Q test is another rapid chromatographic immunoassay that detects serum anti-SARS-CoV-2 antibodies. The nitrocellulose membrane has three test bands, C, G, and M, which are coated with anti-chicken IgY, monoclonal anti-human IgG, and monoclonal anti-human IgM antibodies, respectively. If anti-SARS-CoV-2 antibodies are present, they form immune complexes with gold-labeled recombinant SARS-CoV-2 proteins. The complexes are then captured by the coated isotype-specific antibodies, resulting in violet lines at the corresponding positions. According to the manufacturer, when compared to RT-PCR, the clinical sensitivity and specificity of the STANDARD Q test are 99.03% and 98.65%, respectively.

### 2.4. Statistical Analysis 

The sensitivity, specificity, and percent agreement were calculated based on the results of each test. The results of the RapiSure S1 RBD IgG and nAb tests were compared with those of COVID-19 RT-PCR and PRNT, respectively, to determine false positives or negatives. To determine the strength of agreement between the RapiSure and STANDARD Q tests, Cohen’s kappa (κ) was calculated using MedCalc Software version 20.110 (MedCalc Soft-ware Ltd., Ostend, Belgium). The κ value was interpreted in terms of strength of agreement as follows: <0.20 was poor, 0.20–0.40 was fair, 0.41–0.60 was moderate, 0.61–0.80 was good, and 0.81–1.00 was very good [19]. The *p*-value was calculated using the Chi-squared test in MedCalc, with a *p*-value of less than 0.05 considered statistically significant.

## 3. Results

A total of 78 serum samples confirmed for SARS-CoV-2 infection by COVID-19 RT-PCR comprised 44 males and 34 females, with an average age of 68 years (range, 26–87 years). Two of the 78 samples were negative in the PRNT and reclassified as negative for nAbs, resulting in the 76 PRNT-positive samples. 

When the RapiSure S1 RBD IgG results were compared with those of RT-PCR, the sensitivity was 97.4% (76/78) (95% confidence interval (CI): 91.1–99.3%) and the specificity was 96.7% (118/122) (95% CI: 91.9–98.7%). The results of the RapiSure Neutralizing Ab Test conducted on the PRNT-positive and -negative samples showed a sensitivity of 93.4% (71/76) (95% CI: 85.5–97.2%) and a specificity of 100% (124/124) (95% CI: 97.0–100%). An overall percent agreement between RapiSure and PRNT was 97.5% (95% CI: 94.3% to 99.2%), with a κ value of 0.95. The sensitivity of the RapiSure Neutralizing Ab Test was lower than that of the S1 IgG test; both tests showed high specificity (Table 1).

The sensitivity of the STANDARD Q test compared to the COVID-19 RT-PCR was 95.7% and comparable to that of the RapiSure (97.4%). However, the specificity of STANDARD Q was 90.8%, which was lower than RapiSure (96.7%). Using their corresponding positive and negative results, a comparison between the RapiSure and STANDARD Q tests revealed an overall concordance of 91.5% (95% CI: 86.8% to 94.6%) with a κ value of 0.82, indicating a very good strength of agreement (Table 2). 

The results of the LoD tests using the five positive samples with a PRNT_90_ titer of 1:160 are summarized in Table 3. In the S1 RBD IgG test, one sample remained positive at a 1:32 titer while the other four samples remained positive at a 1:64 titer or higher. Meanwhile, the RapiSure Neutralizing Ab Test showed that one sample remained positive at a 1:16 titer while the others remained positive at a 1:32 titer or higher.

The 76 PRNT-positive samples were divided into three groups according to the number of days after the onset of illness: <7 days (n = 25), 8–14 days (n = 28), and >15 days after symptom onset (n = 23). The sensitivity of the RapiSure test in the <7 days group was 92.6% for S1 RBD IgG and 88.0% for nAbs. In the 8–14 days group, 100% and 92.9% were positive in the S1 RBD IgG and nAb tests, respectively. No false negatives were observed in the >15 days group (Figure 2). 

When the low- and high-titer PRNT-positive groups were applied to the RapiSure test, the low-titer groups of PRNT_50_ and PRNT_90_ showed significantly lower sensitivity (76.9% and 70.0%, respectively) than those of high-titer groups of PRNT_50_ and PRNT_90_ (Table 4). 

## 4. Discussion

Timely detection and clinical decision-making processes for COVID-19 are crucial for infection control and public health management during the ongoing pandemic. The presence of anti-SARS-CoV-2 nAbs helps protect against reinfection by the same strain [5]. Various serological assays to detect anti-SARS-CoV-2 antibodies have been marketed to date [20,21]. Currently, antibody testing is not recommended for confirming immunity after vaccination, according to the interim guidelines by the Centers for Disease Control and Prevention [15]. Moreover, the real-world application of antibody tests is hampered by the ongoing pandemic and the limited capacity of laboratory-based testing. The development of improved tests, including POCT, could substantially accelerate clinical decision-making and help monitor the effectiveness of governmental infection control strategies. Despite these potentials, POCT using lateral flow immunochromatographic assay have concerns about lower sensitivities than laboratory-based serological methods, such as ELISA and chemiluminescence immunoassay [22]. Therefore, test validity and reliability are crucial in POCT: unreliable diagnostic tests can hamper healthcare provision by failing to detect patients with SARS-CoV-2 infection or by incorrectly identifying negative patients as positive [21]. The PRNT is the gold STANDARD for assessing the presence and titer of antibodies in serum samples. However, its demanding requirements and high cost have highlighted the need for the development and validation of anti-SARS-CoV-2 antibody testing in POCT.

Since antibodies against the SARS-CoV-2 RBD are highly correlated with nAbs, anti-RBD antibody assays have been developed to assess post-infection immunity and validate vaccine effectiveness [23,24]. Anti-RBD IgG titers measured by commercial serological assays show a correlation with nAb titers, thus qualifying as a proxy marker of neutralization [25,26]. However, these results were obtained using the RBD of wild-type (WT) SARS-CoV-2, before the emergence of variants of concern (VOCs) [27,28]. 

VOCs, especially Delta and Omicron, raised concerns about their potential for immune escape from vaccine-induced antibodies due to mutations in their S proteins [29]. In addition, there are doubts regarding the performance of previously released anti-RBD antibody assays that use RBD antigens derived from WT SARS-CoV-2. To address these issues, a comparative study was conducted on whether the anti-RBD IgG titers measured using commercially available kits could represent the presence of nAbs against VOCs [30]. According to the study, there was a strong correlation between anti-RBD IgG and nAbs levels against WT and pre-Omicron variants, including Alpha, Beta, and Delta. However, despite a high anti-RBD IgG titer, anti-Omicron nAbs were not observed, indicating no correlation between anti-RBD IgG and nAbs. Omicron subvariants have been reported to escape most of the anti-SARS-CoV-2 nAbs induced by vaccination and infection and elicit substantially lower nAb titers [31]. Reduced sensitivity of serological assays has been observed in assays using recombinant WT S protein for the detection of anti-RBD antibodies, suggesting an underestimation of true antibody titers [32]. 

The RapiSure COVID-19 S1 RBD IgG/Neutralizing Ab Test evaluated in this study detected anti-RBD antibodies and nAbs simultaneously in a single cassette. Two samples were found to be positive by RT-PCR but negative by the PRNT, with a final classification of negative; only one sample was positive for anti-S1 RBD antibodies. Both samples had less than one symptomatic day after the onset of illness. The sensitivity of immunoassays can be affected by the viral load and sample collection date, which can both affect circulating antibody levels [33]. Antibody titers in the recovery period (>7 days after diagnosis) are significantly higher than those in the acute phase (≤7 days) [6]. Additionally, the induction of nAbs in the acute phase has been reported to vary depending on disease severity [34]. The results of the RapiSure nAb test in the two samples were consistent with those of the PRNT; the antibody test results were consistent.

Serological detection of anti-SARS-CoV-2 antibodies usually begins at the end of the first week of infection, and peak nAb detection is reached after 3–4 weeks of illness [8,35,36]. Serum collected >15 days after the onset of illness showed 100% positivity in both the RapiSure IgG and nAb tests. Meanwhile, of the samples collected 8–14 days after the onset of illness, 100% positivity was observed in the RapiSure S1 RBD IgG compared to the results of PRNT, with two false negative samples in the RapiSure nAb test. These discordant results between S1 RBD IgG and nAb test did not indicate the possibility of Omicron infection, since South Korea’s first Omicron cases were identified on 25 November 2021 [37]. However, the simultaneous measurement of anti-S1 RBD antibodies and nAb may be very useful in the detailed descriptions of immune status, especially in cases of VOC infection.

There are several limitations to this study. First, sample collection occurred before Omicron became the predominant variant, hindering the relevance of the results to this variant. Second, the RapiSure test results were interpreted with the naked eye rather than any specific reading devices; therefore, the interpretation of weak test reactions, often shown as faint lines, may not be objective. However, the performance of RapiSure showed better sensitivity (97.4% vs. 95.7%) and specificity (96.7% vs. 90.8%) than the previously marketed STANDARD Q test when compared to COVID-19 RT-PCR. The ability to detect nAb was comparable to the PRNT, with an overall percent agreement of 97.5%. Nonetheless, the qualitative nature of the RapiSure test, including reduced sensitivity at an early stage (<7 days), may hamper the proper diagnosis for the level of protection. Finally, a few false positives (RapiSure: four cases, STANDARD Q test: three cases) were observed, but the causative agent that can cause cross-reactivity remained unclear. According to the package insert of RapiSure, there was no cross-reactivity with anti-influenza A virus, anti-influenza B virus, anti-respiratory syncytial virus, anti-adenovirus, hepatitis B surface antigen, anti-syphilis, anti-helicobacter pylori, anti-human immunodeficiency virus, anti-hepatitis C virus, and human anti-mouse antibody positive specimens. However, the possibility of false reactivity due to coinfection with another pathogen cannot be ruled out. 

Previously reported sensitivity and specificity of RapiSure were 96.8% and 97.7% for the S1 RBD IgG and 92.2% and 100.0% for the nAb test, respectively, which were similar to this study [38]. The limitation of POCT includes lowered and large ranges of sensitivity reported by the different POCT tests. According to the final WHO SARS-CoV-2 serology test kit evaluation that evaluated 26 rapid diagnostic tests (RDTs) using lateral flow assay, the sensitivity of IgG ranged from 77.4% to 100.0%, and specificity ranged from 81.0 to 99.0%, respectively [39]. Considering the similarity of sensitivity and specificity calculated on different populations and the performance of other RDTs, the RapiSure showed an acceptable performance. 

Compared to anti-RBD antibodies, the status of nAbs was a better indication of immunity against variants. However, commercially available antibody assays, which are widely used, typically measure anti-RBD IgG levels. Since the Omicron variant is currently the globally dominant strain [27], the anti-RBD IgG test alone might provide unreliable information about the immunity status following infection or vaccination.

## 5. Conclusions

In conclusion, the RapiSure (EDGC) COVID-19 S1 RBD IgG/Neutralizing Ab Test showed reliable sensitivity and specificity as a lateral flow test, and the simultaneous detection of anti-RBD antibodies and nAbs is of advantage. This assay has the potential to be conveniently used in the Omicron era when the anti-RBD IgG test alone might be unreliable for detecting the level of protection.

## Figures and Tables

**Figure 1 diagnostics-13-00643-f001:**
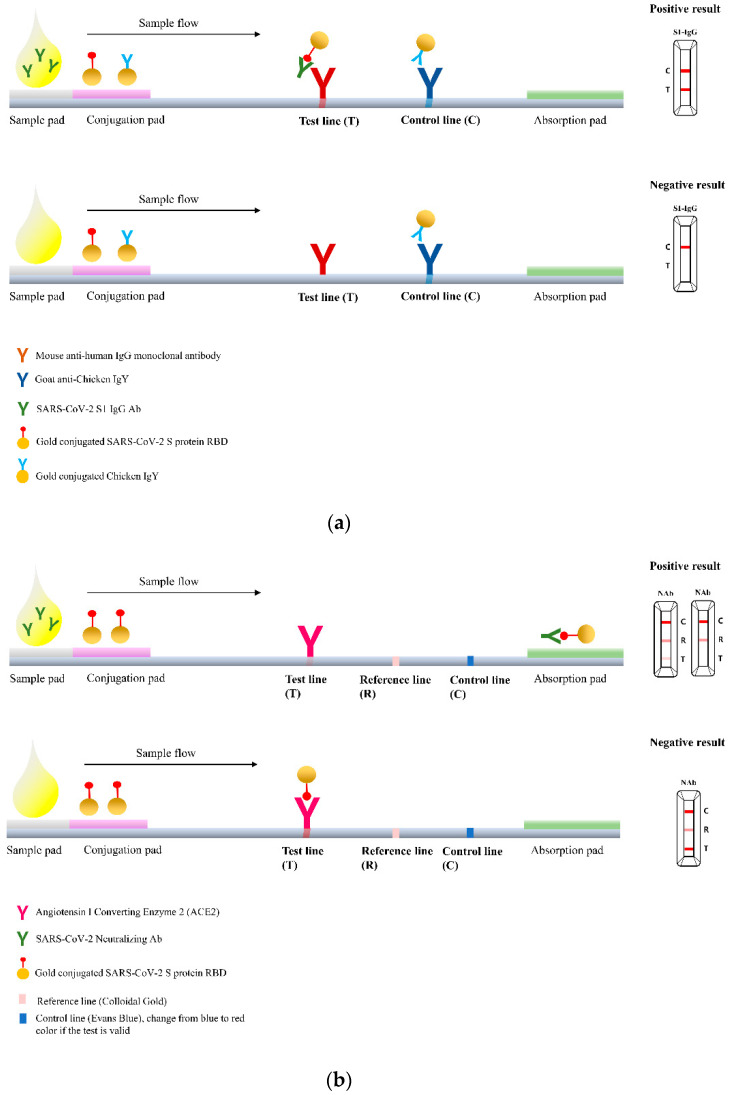
Schematic diagram and images of the RapiSure COVID-19 S1 RBD IgG/Neutralizing Ab Test. (**a**) S1 RBD IgG; (**b**) neutralizing antibody; and (**c**) images of the RapiSure COVID-19 S1 RBD IgG/Neutralizing Ab Test using representative serum samples with positive, weakly positive, and negative results from left to right.

**Figure 2 diagnostics-13-00643-f002:**
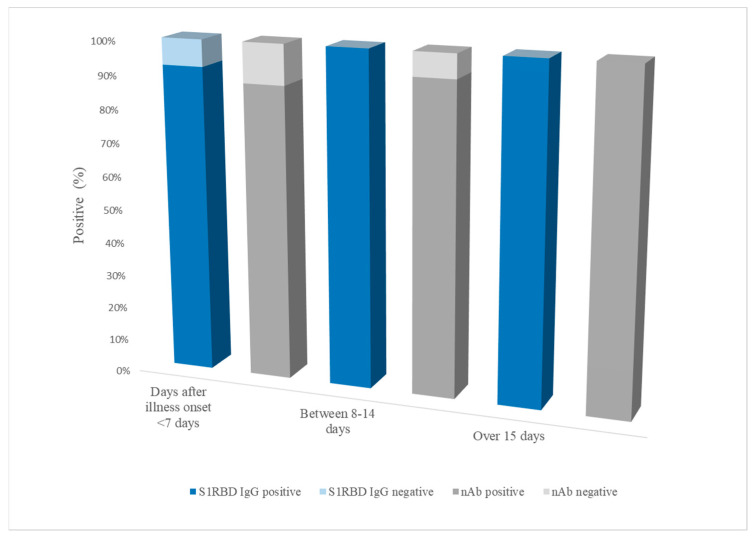
Sensitivity (%) of the RapiSure COVID-19 S1 RBD IgG/Neutralizing Ab Test in three groups: <7 days, 8–14 days, and >15 days after onset of illness. Blue bars represent the results for S1 RBD IgG; gray bars represent the results for nAbs, of which each upper light-colored bar represents a false negative result.

**Table 1 diagnostics-13-00643-t001:** Performance of RapiSure COVID-19 S1 RBD IgG/Neutralizing Ab Test compared to the results of COVID-19 RT-PCR and PRNT_50_/PRNT_90._ The RapiSure S1 RBD IgG test results were compared with those of RT-PCR, and the Neutralizing Ab test results were compared with those of PRNT, respectively.

		RT-PCR ^a^		Sensitivity (95% CI ^b^)	Specificity (95% CI)	PRNT ^c^_50_ and PRNT_90_		Sensitivity (95% CI)	Specificity (95% CI)	Percent Agreement (95% CI)	Cohen’s Kappa (95% CI)
		Positive	Negative			Positive	Negative				
**RapiSure**	Positive	76	4	97.4% (91.1–99.3%)	96.7% (91.9–98.7%)	71	0	93.4% (85.5–97.2%)	100% (97.0–100%)	97.5% (94.3–99.2%)	0.95 (0.90–0.99)
	Negative	2	118			5	124				

^a^ COVID-19 RT-PCR: STANDARD™ M nCoV Real-Time Detection kit or AllplexTM 2019-nCoV assay; ^b^ CI: Confidence interval; ^c^ Plaque reduction neutralization test.

**Table 2 diagnostics-13-00643-t002:** Comparison between the RapiSure COVID-19 S1 RBD IgG test results and the IgG test results obtained using STANDARD Q COVID-19 IgM/IgG Plus. The results of the STANDARD Q IgG test were also compared to those of COVID-19 RT-PCR.

		STANDARD Q IgG		Sensitivity (95% CI)	Specificity (95% CI)	Percent Agreement ^a^ (95% CI)			Cohen’s Kappa (95% CI)
		Positive	Negative			Positive	Negative	Overall	
RapiSure S1 RBD IgG	Positive	66	14			95.7% (88.0–98.5%)	89.3% (82.9–93.5%)	91.5% (86.8–94.6%)	0.82 (0.73–0.90)
	Negative	3	117						
RT-PCR	Positive	66	12	95.7% (87.8–99.1%)	90.8% (84.5–95.2%)			92.5% (87.9–95.7%)	0.84 (0.76–0.92)
	Negative	3	119						

^a^ Calculated as positive = (RapiSure and STANDARD Q both positive/STANDARD Q positive) × 100, negative = (RapiSure and STANDARD Q both negative/STANDARD Q negative) × 100, overall = (RapiSure and STANDARD Q both positive + RapiSure and STANDARD Q both negative)/total × 100.

**Table 3 diagnostics-13-00643-t003:** Evaluation of the limit of detection of the RapiSure COVID-19 S1 IgG/Neutralizing Ab Test using positive samples with a PRNT_90_ titer of 1:160. The gray-shaded boxes indicated their lowest dilution factor.

	Sample 1		Sample 2		Sample 3		Sample 4		Sample 5	
Dilution Factor	S1-IgG	nAb	S1-IgG	nAb	S1-IgG	nAb	S1-IgG	nAb	S1-IgG	nAb
1:2	+	+	+	+	+	+	+	+	+	+
1:4	+	+	+	+	+	+	+	+	+	+
1:8	+	+	+	+	+	+	+	+	+	+
1:16	+	+	+	+	+	+	+	+	+	+
1:32	+	+	+	+	+	+	+	+	+	−
1:64	+	−	+	+	+	+	+	+	−	−
1:128	−	N/T ^a^	+	−	+	+	+	+	N/T	N/T
1:256	−	N/T	+	−	−	−	−	−	N/T	N/T

^a^ N/T: not tested.

**Table 4 diagnostics-13-00643-t004:** Results of the RapiSure COVID-19 Neutralizing Ab Test according to low- and high-titer groups in the PRNT assay.

	Categorization	Titer	Numbers of Positive/Total Samples	RapiSure nAb Results		Sensitivity (95% CI)	*p*-Value ^a^
				Positive	Negative		
PRNT_50_	Low titer	1:20, 1:40, 1:80	13/76	10	3	76.9% (49.7–91.8)	0.0090
	High titer	>1:80	63/76	61	2	96.8% (89.1–99.1)	
PRNT_90_	Low titer	1:10, 1:20, 1:40	10/76	7	3	70.0% (39.7–89.2)	0.0014
	High titer	>1:40	66/76	64	2	97.0% (89.3–99.2)	

^a^ Calculated using Chi-squared test, a *p*-value less than 0.05 indicates a statistically significant difference.

## Data Availability

The data supporting the findings of this study are available from the corresponding author upon reasonable request.

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
