# Peer review of "Performance Evaluation of RapiSure (EDGC) COVID-19 S1 RBD IgG/Neutralizing Ab Test for the Rapid Detection of SARS-CoV-2 Antibodies"

_diagnostics, 2023, doi:10.3390/diagnostics13040643_

Round 1
Reviewer 1 Report
In this manuscript ‘Performance Evaluation of RapiSure (EDGC) COVID-19 S1 RBD IgG/Neutralizing Ab Test for the Rapid Detection of SARS-CoV-2 Antibodies’, Kim et al describes the performance characteristics of a commercial POCT Ab test using pre-Omicron clinical specimens.
# General comment;
Overall, the test performance is well-described using PRNT/STANDARD Q reference testing platforms. The results seem promising and support the advantage of the use of this POCT especially in remote areas with limited testing resource. Manuscript is concise and well-written.
# Specific major comments;
* Judging from sensitivity/specificity data, the POCT performance seems promising. In fact, however, an in-house PRNT was used as reference standard, the performance (limit of detection (LOD) etc.) of which is not necessarily made clear to external readers. In such a case, the use of standard sera (i.e. WHO IS sera) will be highly helpful in increasing cross-assay comparability and the critical assessment of study results by the broad international readership. Have the authors tested their PRNT protocol using WHO standards? Is LOD of the assay discussible using WHO nAb IU (international units)?
* Can the authors specify the exact strain for which the assay antigens were designed (for PRNT, RapiSure, and STANDARD Q)? Also, the circulating variants during the participant recruitment phase (from December 2020 to September 2021) are to be noted.
* Most importantly, as far as I understand the PRNT/STANDARD Q test assay designs, I am afraid that the comment "there was a strong correlation between nAbs against WT and pre-Omicron variants” is not fully supported by comparing RapiSure results with PRNT/STANDARD Q test results, and is not accurately summarizing the study findings. Instead, does this simply mean that “the assay performance was satisfying even when testing pre-Omicron variants”? A further explanation or a better description is warranted.
# More minor comments;
* In Table 1 in the present form, it is difficult to tell without the help of the text that S1 Ab results and nAb results were compared with RT-PCR and PRNT results, respectively. A footnote or comment in the legend may assist the understanding of readers. The same applies to Table 2. Which of the S1 Ab or nAb test in the RapiSure platform was used is not apparent from the Table itself and/or its legend.
* Page 2, lines 69–70; Was 'symptomatic day after the onset of illness’ recorded to serve as index of severity? The purpose of this sentence is not clear. A brief description of donors demographics (proportion of each symptom severity class, age distribution, antibody titer distribution (described in WHO standard units of BAU/IU if possible)) will be highly appreciated.
* Page 7, lines 208–9; The authors have commented on evaluating protective immunity of an individual through the testing of nAbs. The qualitative nature of RapiSure nAb test should be noted as a limitation, hampering the inference of the degree of protection solely from RapiSure testing results.
* Page 9, liness 279-81; I agree that 'the simultaneous detection of anti-RBD antibodies and nAbs’ is definitely an advantage of the RapiSure assay platform. How advantageous this shall perform in the post-Omicron era remains questionable. Can the authors further comment on this "great advantage” they propose?
Author Response
Response to Reviewer 1 Comments
Thank you for giving me the opportunity to submit a revised draft of my manuscript to Diagnostics. We appreciate the time and effort that the reviewers have dedicated to providing your valuable feedback on my manuscript. The modified sentences were highlighted in the manuscript.
# Specific major comments;
Point 1: Judging from sensitivity/specificity data, the POCT performance seems promising. In fact, however, an in-house PRNT was used as reference standard, the performance (limit of detection (LOD) etc.) of which is not necessarily made clear to external readers. In such a case, the use of standard sera (i.e. WHO IS sera) will be highly helpful in increasing cross-assay comparability and the critical assessment of study results by the broad international readership. Have the authors tested their PRNT protocol using WHO standards? Is LOD of the assay discussible using WHO nAb IU (international units)?
Response 1: The PRNT was performed at the Department of Microbiology of Korea University where a bio-safety level three facility is available. The method for PRNT followed the previously published standardized laboratory method by Food and Drug Administration (Vaccine (2007) 26, 59—66). Including the study that we had referenced in the manuscript (Yonsei Med J 2021 Jul;62(7):584-592), the results of PRNT performed in this institution using the same procedure were used in other studies as the reference method (Diagnostics 2021, 11(12), 2193; https://doi.org/10.3390/diagnostics11122193, Diagnostics 2022, 12(8), 1924; https://doi.org/10.3390/diagnostics12081924). We also used PRNT results using the same methods and procedures at the same institution as in the studies mentioned above.
In brief, the SARS-CoV-2 of S clade (BetaCoV/Korea/KCDC03/2020, NCCP 43326) was used for neutralization activity analysis. The viral genomic RNA of SARS-CoV-2 (BetaCoV/Korea/KCDC03/2020) was obtained from the National Cell Collection for Pathogens (NCCP) of South Korea. Virus-inoculated Vero cell plates (NEST Scientific, SPL Life Sciences, Pochen, Korea were inoculated with the serum-virus mixtures.
Point 2: Can the authors specify the exact strain for which the assay antigens were designed (for PRNT, RapiSure, and STANDARD Q)? Also, the circulating variants during the participant recruitment phase (from December 2020 to September 2021) are to be noted.
Response 2: The strain used in PRNT was the wild-type (BetaCoV/Korea/KCDC03/2020), obtained from the National Cell Collection for Pathogens (NCCP) of South Korea. According to data from Korea Disease Control and Prevention Agency (KDCA), the dominant COVID-19 variant in South Korea was the Alpha variant until April 2021, with detection rate of 13.7%. Delta variant was first confirmed in April 2021 (0.1%), and became the dominant species in July with 59.6%. When inferred from the variant information in Korea, RapiSure and STANDARD Q are assumed to use wild-type stains. Still, it may be inaccurate as there are no available data from the manufacturers to the best of my knowledge. Strain analysis was not performed on the collected samples in this study. However, since Omicron first occurred in Korea on November 25, 2021, all our samples collected before then were considered non-Omicron variants.
Point 3: Most importantly, as far as I understand the PRNT/STANDARD Q test assay designs, I am afraid that the comment "there was a strong correlation between nAbs against WT and pre-Omicron variants” is not fully supported by comparing RapiSure results with PRNT/STANDARD Q test results, and is not accurately summarizing the study findings. Instead, does this simply mean that “the assay performance was satisfying even when testing pre-Omicron variants”? A further explanation or a better description is warranted.
Response 3: I apologize for the misunderstanding caused by the missing part of the sentence. There were missing words in the comment "there was a strong correlation between nAbs against WT and pre-Omicron variants”. I corrected the sentence to “there was a strong correlation between anti-RBD IgG and nAbs levels against WT and pre-Omicron variants, including Alpha, Beta, and Delta” (Page 8, Line 247-248). According to the study conducted by Takheaw N et al. (Diagnostics 2022, 12, 1315. https://doi.org/10.3390/ diagnostics12061315), anti-RBD IgG levels did not correlate with nAb levels against Omicron with a negative correlation (Spearman’s correlation coefficient of -0.514), revealing no correlation between anti-RBD IgG and nAbs against Omicron. On the other hand, the anti-RBD IgG levels had a strong correlation with nAbs against WT, Alpha, Beta, and Delta variants based on Spearman’s correlation coefficient analysis. Therefore, I intended to emphasize the usability of the RapiSure test in the context of simultaneous detection of anti-RBD and nAb since the positivity of anti-RBD IgG and nAb may reveal discrepancies depending on the type of variants.
# More minor comments;
Point 4: In Table 1 in the present form, it is difficult to tell without the help of the text that S1 Ab results and nAb results were compared with RT-PCR and PRNT results, respectively. A footnote or comment in the legend may assist the understanding of readers. The same applies to Table 2. Which of the S1 Ab or nAb test in the RapiSure platform was used is not apparent from the Table itself and/or its legend.
Response 4: Thank you for pointing this out. I added a supplementary explanation in the legend of Table 1 and 2. The results of RapiSure COVID-19 S1 RBD IgG test and the IgG test results obtained using STANDARD Q COVID-19 IgM/IgG Plus were compared for analysis.
Point 5: Page 2, lines 69–70; Was 'symptomatic day after the onset of illness’ recorded to serve as index of severity? The purpose of this sentence is not clear. A brief description of donors demographics (proportion of each symptom severity class, age distribution, antibody titer distribution (described in WHO standard units of BAU/IU if possible)) will be highly appreciated.
Response 5: The number of days after illness onset was recorded for the performance evaluation of the RapiSure test to detect developing antibodies that change according to the number of days after onset, but not for the index of severity. The proportion of gender and average age were retrieved from the sample data, and the information was added in the results. However, since this study focused on the performance evaluation of the RapiSure, we did not investigate the severity of the symptoms through a preliminary survey for objective analysis. Therefore, it was difficult to determine the proportion of each symptom severity class. In addition, we could not perform additional qualitative measurements of antibodies (e.g., ELISA) due to the insufficient volume of samples.
Point 6: Page 7, lines 208–9; The authors have commented on evaluating protective immunity of an individual through the testing of nAbs. The qualitative nature of RapiSure nAb test should be noted as a limitation, hampering the inference of the degree of protection solely from RapiSure testing results.
Response 6: I agree with this and have incorporated your suggestion in the discussion section (Page 9, Line 283-285) as one of the limitations of the study.
Point 7: Page 9, liness 279-81; I agree that 'the simultaneous detection of anti-RBD antibodies and nAbs’ is definitely an advantage of the RapiSure assay platform. How advantageous this shall perform in the post-Omicron era remains questionable. Can the authors further comment on this "great advantage” they propose?
Response 7: Thank you for your suggestion. I have, accordingly, revised the conclusion section, with deleting the word “great”.
Currently, most of the commercially available antibody assays measure only anti-RBD IgG levels. Since a high correlation was observed between the RBD antibodies and nAbs, anti-RBD IgG results alone could be evidence of protection before the emergence of Omicron variants. The Omicron variants revealed multiple mutations in the S protein. Takheaw N et al. (Diagnostics 2022, 12, 1315. https://doi.org/10.3390/ diagnostics12061315) have demonstrated that among vaccinated individuals with high anti-RBD IgG levels, none of them had detectable nAbs against Omicron. Their results indicate that anti-RBD IgG levels cannot be used as a predictor for the presence of nAbs against the Omicron variant.
Because of these characteristics of the Omicron variant, I considered the RapiSure test would be more useful than other assays that detect only RBD IgG when Omicron variants are dominant. Unfortunately, the results of Omicron infection on RapiSure could not be confirmed due to the selection period of the sample in this study. As all our samples were collected from individuals infected with SARS-CoV-2 from December 2020 to September 2021, it is assumed that no samples were obtained from individuals infected with Omicron variants. Assuming the Omicron infection, the results of RapiSure could be positive for RBD IgG, but nAb may be negative. I think the further evaluation of Omicron-infected cases would be interesting.
Reviewer 2 Report
Dear Editor,
I appreciate you giving me the opportunity to review a manuscript on the topic „Performance Evaluation of RapiSure (EDGC) COVID-19 S1 RBD IgG/Neutralizing Ab Test for the Rapid Detection of SARS-CoV-2 Antibodies” by Kim HN et al. the characteristics of the test RapiSure (EDGC)COVID-19 S1 RBD IgG/Neutralizing Ab test were compared with Standard QCOVID-19 IgM/IgG Plus test in the paper.
In the introductory part of the paper, a hundred or so tests are described, I propose to briefly describe the characteristics of this test, the STANDARD Q COVID-19 IgM/IgG Plus test, with which the examined test was compared, as well as highlight the shortcomings of the tests applied so far. It lacks general, it is just a test of the STANDARD Q COVID-19 IgM/IgG Plus test taken as a comparator.
Materials method description is about sample collection, characteristics Rpi Sure COVID-19 S1 RBD IgG/Neutralizing Ab test, description comparison of the Rpi Sure test with the PRNT and the STANDARD Q COVID-19 IgM IgG plus test. It should be described in more detail how false positives and false negatives were determined.
The discussion lacks the experiences of other authors with the examined test, what are the disadvantages of the tested test and what the sensitivity compared to other strains
Author Response
Response to Reviewer 2 Comments
Thank you for giving me the opportunity to submit a revised draft of my manuscript to Diagnostics. We appreciate the time and effort that the reviewers have dedicated to providing your valuable feedback on my manuscript. The modified sentences were highlighted in the manuscript.
Point 1: In the introductory part of the paper, a hundred or so tests are described, I propose to briefly describe the characteristics of this test, the STANDARD Q COVID-19 IgM/IgG Plus test, with which the examined test was compared, as well as highlight the shortcomings of the tests applied so far. It lacks general, it is just a test of the STANDARD Q COVID-19 IgM/IgG Plus test taken as a comparator.
Response 1: Thank you for pointing this out. I added the sentence in Page 2, Line 59-60. STANDARD Q COVID-19 IgM/IgG Plus Test (Standard Q) by SD Biosensor was the first diagnostic devices to detect SARS-CoV-2 antibodies approved by Ministry of Foods and Drug Safety in Korea on 6 November 2020. Also, the Standard Q is one of the evaluated 35 products in the WHO SARS-CoV-2 serology evaluation (Final WHO SARS-CoV-2 serology test kit evaluation results, published on 21 July 2022). According to the evaluation report, only five tests reported no false reactivity for the 55 samples with potentially cross-reacting substances and Standard Q was one of them. As such, the Standard Q was used as a comparative device because of its proven performance. The brief characteristics of Standard Q test were described in the method section (Page 4, Line 129-137).
Point 2: Materials method description is about sample collection, characteristics Rpi Sure COVID-19 S1 RBD IgG/Neutralizing Ab test, description comparison of the Rpi Sure test with the PRNT and the STANDARD Q COVID-19 IgM IgG plus test. It should be described in more detail how false positives and false negatives were determined.
Response 2: I agree with this comment. Therefore, I have added the sentence in the materials and methods section (Page 5, Line 140-142). The results of RapiSure S1 RBD IgG and nAb test were compared with those of COVID-19 RT-PCR and PRNT, respectively, to determine false positives or negatives.
Point 3: The discussion lacks the experiences of other authors with the examined test, what are the disadvantages of the tested test and what the sensitivity compared to other strains
Response 3: I agree with this suggestion and have added sentences to emphasize this point, including the content about the false reactions and the results of the previous study in the discussion section (Page 8, Line 225-228, Page 9, Line 285-301).

Round 2
Reviewer 1 Report
The authors have responded sincerely to all my comments.
All amendments made to the original version have aided in clarifying the methods followed, the results obtained, and the limitations faced, altogether leading to provision of a better opportunity to receive appraisal from its readerships.